# Dietary Micronutrients and Risk of Chronic Kidney Disease: A Cohort Study with 12 Year Follow-Up

**DOI:** 10.3390/nu13051517

**Published:** 2021-04-30

**Authors:** Juyeon Lee, Kook-Hwan Oh, Sue-Kyung Park

**Affiliations:** 1Department of Preventive Medicine, College of Medicine, Seoul National University, 103 Daehakro, Jongnogu, Seoul 03080, Korea; juyeon87@snu.ac.kr; 2Department of Biomedical Science, College of Medicine, Seoul National University, 103 Daehakro, Jongnogu, Seoul 03080, Korea; 3Department Cancer Institution, Seoul National University, 103 Daehakro, Jongnogu, Seoul 03080, Korea; 4Division of Nephrology, Department of Internal Medicine, Seoul National University Hospital, 103 Daehakro, Jongnogu, Seoul 03080, Korea; khoh@snu.ac.kr; 5Integrated Major in Innovative Medical Science, College of Medicine, Seoul National University, 103 Daehakro, Jongnogu, Seoul 03080, Korea

**Keywords:** micronutrient, minerals intake, vitamins intake, chronic kidney disease

## Abstract

We investigated the association between dietary micronutrient intakes and the risk of chronic kidney disease (CKD) in the Ansan-Ansung study of the Korean Genome and Epidemiologic Study (KoGES), a population-based prospective cohort study. Of 9079 cohort participants with a baseline estimate glomerular filtration rate (eGFR) ≥60 mL/min/1.73 m^2^ and a urine albumin to creatinine ratio (UACR) <300 mg/g and who were not diagnosed with CKD, we ascertained 1392 new CKD cases over 12 year follow-up periods. The risk of CKD according to dietary micronutrient intakes was presented using hazard ratios (HRs) and 95% confidence intervals (95% CIs) in a full multivariable Cox proportional hazard models, adjusted for multiple micronutrients and important clinico-epidemiological risk factors. Low dietary intakes of phosphorus (<400 mg/day), vitamin B2 (<0.7 mg/day) and high dietary intake of vitamin B6 (≥1.6 mg/day) and C (≥100 mg/day) were associated with an increased risk of CKD stage 3B and over, compared with the intake at recommended levels (HR = 6.78 [95%CI = 2.18–21.11]; HR = 2.90 [95%CI = 1.01–8.33]; HR = 2.71 [95%CI = 1.26–5.81]; HR = 1.83 [95%CI = 1.00–3.33], respectively). In the restricted population, excluding new CKD cases defined within 2 years, an additional association with low folate levels (<100 µg/day) in higher risk of CKD stage 3B and over was observed (HR = 6.72 [95%CI = 1.40–32.16]). None of the micronutrients showed a significant association with the risk of developing CKD stage 3A. Adequate intake of micronutrients may lower the risk of CKD stage 3B and over, suggesting that dietary guidelines are needed in the general population to prevent CKD.

## 1. Introduction

Chronic kidney disease (CKD) is defined either as at least 3 months of a reduced estimated glomerular filtration rate (eGFR) <60 mL/min/1.73^2^ or kidney damage, or as the presence of persistent albuminuria [1]; it is considered a significant public health problem with an increasing prevalence. According to different countries, including the United States (U.S.), CKD affects 8–16% of adults worldwide [2]. In the Korean population, the prevalence of CKD is 8.2% [3]. 

Early diagnosis of CKD is important for reducing mortality, improving treatment outcomes, and managing complications such as cardiovascular disease (CVD) [4]. Potential risk factors for CKD include older age, diabetes mellitus, hypertension, obesity, alcohol consumption, smoking, and poor dietary habits. Dietary intake, as a modifiable risk factor, plays an important role in CKD development [5,6]. It has been reported in three studies that higher intake of certain antioxidant vitamins or healthy diet (e.g., Mediterranean diet) may improve kidney function or reduce CKD [7,8,9]. Despite epidemiologic evidence, the associations of dietary minerals and vitamins with the development of CKD have not been universally defined, and the results for each micronutrient have been inconsistent because most studies either used the cross-sectional study design or focused on patients on dialysis [7,10]. In particular, it is not clear how much to eat to prevent CKD, and what is the appropriate intake, which is defined as the minimum risk for developing CKD.

In the present study, we investigated the overall and subgroup risk of CKD stage 3A and stage 3B and over according to dietary micronutrient intake using the 12 year follow-up data of the Ansan-Ansung cohort from the Korea Genome and Epidemiologic Study (KoGES). The subgroup was classified into diabetic CKD, hypertensive CKD, and other CKD by underlying causes (subjects who had diabetes at the time of baseline measurement and subsequently developed CKD were defined as diabetic CKD; subjects who had hypertension (HTN) at the time of baseline measurement and subsequently developed CKD were defined as hypertensive CKD). 

## 2. Materials and Methods

### 2.1. Study Population

The Ansan-Ansung study, a prospective population-based cohort study, is part of the KoGES. Further details of the design and methodologies of the Ansan-Ansung study (Ansan: representing the urban community, Ansung: representing the rural community) and the KoGES have been presented elsewhere [11]. This research is an ongoing prospective cohort study involving biennial follow-ups. At baseline, 10,038 participants (aged 40–69 years), including 5020 from Ansan and 5018 from Ansung, were recruited between 2001 and 2002 through the cluster sampling method. Subjects were followed up until the 6th regular follow-up survey conducted between 2013 and 2014. At each visit, informed written consent was obtained from all subjects. Among the 10,030 subjects, we excluded those with a low eGFR (<60 mL/min/1.73 m^2^) at the cohort enrollment, as calculated using the Chronic Kidney Disease Epidemiology Collaboration (CKD-EPI) formula, those with a urine albumin to creatinine ratio ≥300 mg/g, those with a known history of kidney disease (*n* = 485), those without information on serum creatinine levels (*n* = 2), those who did not complete a food frequency questionnaire (FFQ) (*n* = 307), and those with a history of cardiovascular disease (*n* = 157) (Figure 1). The final analysis was therefore conducted on 9079 participants, of whom 1392 developed CKD (eGFR < 60 mL/min/1.73^2^) and 7687 did not (eGFR ≥ 60 mL/min/1.73^2^). In addition, we performed sensitivity analyses restricted to a eGFR <60 mL/min/1.73^2^ and CKD cases diagnosed within 2 years after study entry (8901 participants, of whom 1115 developed CKD and 7786 did not develop CKD). 

This study was conducted according to the guidelines established by the Declaration of Helsinki. Written informed consent was obtained from all subjects, and this study was approved by the National Biobank of Korea and the Centers for Disease Control and Prevention, Republic of Korea. The study protocols were approved by the institutional review board (IRB) of Seoul National University Hospital. 

### 2.2. Ascertainment of New CKD Cases over 12 Year Follow-Up 

CKD was defined as a ‘eGFR < 60 mL/min/1.73 m^2^’, in compliance with the National Kidney Foundation Kidney Disease Outcomes Quality Initiative (NKF KDOQI) clinical practice guideline [12]. A new case of CKD was defined as a eGFR decline to <60 mL/min/1.73 m^2^ over a follow-up period ranging from 2 to up to 12 years among cohort members with an eGFR greater than 60 mL/min/1.73 m^2^ at the time of baseline. The eGFR was calculated using the CKD-EPI equation [13]. The CKD-EPI equation is shown as Equation (1).
(1)$$\mathrm{eGFR}=141\times \min{\left(\frac{Scr}{k}\right)}^{\alpha }\times \max{\left(\frac{Scr}{k}\right)}^{-1.209}\times 0.993^{Age}\times 1.018\left[if\;female\right]$$

*Scr* = serum creatinine (mg/dL); *k* = 0.7 if female; 0.9 if male; *α* = −0.329 if female; −0.411 if male.

min = the minimum of *Scr*/*k* or 1; max = the maximum of *Scr*/*k* or 1.

Of the 9079 cohort population, the number of new cases of CKD stage 3A was 1239, of which 1140 were completed in stage 3A over 12 year follow-up, and 99 were deteriorated with a eGFR from stage 3A to stage 3B, 4, or 5. There were 153 new patients ascertained as ‘CKD stage 3B or higher’, of which 99 were deteriorated from CKD stage 3A as previously mentioned, and the remaining 54 were new cases who were directly ascertained as stages 3B, 4, or 5 without being ascertained as stage 3A during biennial follow-up. Since biennially repeated measurements/follow-up was conducted in this cohort study, the 54 patients can be defined as patients who rapidly deteriorated during follow-up.

The patients ascertained as CKD stage 3A were considered as early CKD patients without any eGFR deterioration and some of them might be transient or temporary CKDs due to biennial follow-up setting of community-based cohort study. However, since patients ascertained as ‘stage 3B or higher’ were those who worsened from stage 3A during follow-up or had experienced rapid eGFR deterioration during biennial follow-up, these may be considered as ‘definite CKD cases’ on the setting of community-based cohort study. Moreover, since the risk of mortality was different between two patient group of CKD stage 3A and 3B and over [14,15], two patient groups were defined as the group with different clinical characteristics. Thus, based on this consideration, we classified CKD into three groups: non-CKD, defined as a eGFR ≥60 mL/min/1.73 m^2^; CKD stage 3A, 45 mL/min/1.73 m^2^ < eGFR < 60 mL/min/1.73 m^2^; and stage 3B and over, a eGFR <45 mL/min/1.73 m^2^.

### 2.3. Clinical and Laboratory Measurements

Questionnaires surveys, clinical investigations, and physical examinations were performed during the baseline and follow-up assessments. The subjects were questioned by well-trained interviewers regarding their sociodemographic status, lifestyle (smoking, alcohol consumption, and physical activity), anthropometric measurements, and personal and family medical history. All samples were immediately sent to the Korean National Biobank; the reliability of each biomarker analysis has been previously published [10]. Blood and urine samples were collected in a serum separator tube (SST), and a two-ethylene ediaminetetra acetic acid (EDTA) tube, along with a 10 mL midstream urine sample. Both serum and plasma were prepared and aliquoted, and 100–800 µg of blood DNA and 6–10 vials (300–500 µL per vial) were also prepared. For each tube, a two-dimensional barcode was used as a label. Serum creatinine was assayed using the Jaffe method with a HITACHI Automatic Analyzer 7600 (Hitachi, Tokyo, Japan) and ADVIA 1650 Auto Analyzer (Siemens, Washington, DC, USA) [16]. 

### 2.4. Dietary Assessment

The daily nutrient intakes were assessed using a validated, semi-quantitative FFQ developed for the KoGES [16]. All subjects were asked to estimate their average serving of 106 food items and their frequency of consumption. Dietary intakes per day were calculated by combining the serving frequency and portion per unit for each food item with the average amount per serving. We divided the dietary intake of 6 minerals (calcium, phosphorus, sodium, potassium, iron, and zinc) and 10 vitamins (vitamin A, retinol, vitamin B1, vitamin B2, niacin, vitamin B6, folate, vitamin C, carotene, and vitamin E) into recommended values and used them for further analyses. The recommended dietary allowance (RDA), adequate intake (AI), or National Kidney Foundation guideline values of each mineral and vitamin for individuals aged 51–70 years were designated as the reference category. The reference values of each mineral were as follows: calcium (≥600 mg/day), phosphorus (700–1200 mg/day), sodium (2000–2999 mg/day), potassium (≥3400 mg/day), iron (7–10 mg/day), and zinc (8–11 mg/day) (Tables 1 and 2); those of each vitamin were as follows: vitamin A (≥700 R.E/day), retinol (≥100 µg/day), carotene (≥3400 µg/day), vitamin E (≥11 mg/day), vitamin B1 (0.9–1.2 mg/day), vitamin B2 (0.9–1.2 mg/day), niacin (14–19 mg/day), vitamin B9 (≥400 µg/day), and vitamin C (75–100 mg/day) (Appendix A). 

### 2.5. Statistical Analyses 

To compare the general characteristics of the CKD (eGFR < 60 mL/min/1.73 m^2^) and non-CKD (eGFR ≥ 60 mL/min/1.73 m^2^) groups, we conducted an independent t-test for continuous variables and the chi-square test for categorical variables. In order to overcome the reverse causation in the association between dietary factors and CKD incidence to some extent, a ‘restricted population’ group was additionally formed after excluding newly defined CKD cases within 2 years. 

We examined the associations of dietary minerals and vitamins with CKD stage 3A and CKD stage 3B and over using the Cox proportional hazard regression models in the entire population and in the restricted population, respectively. First, to analyze the association with a single micronutrient on the risk of CKD, a Cox regression model was constructed by controlling the following 10 clinico-epidemiological factors, which were statistically significant in each univariable analysis: baseline age, baseline eGFR, sex, body mass index (BMI), physical activity, cigarette smoking, alcohol consumption, total cholesterol levels, the urine albumin to creatinine ratio, and uric acid in urine (Appendix A). Additionally, two factors, such as total calories and protein intake were additionally included in the multivariable model for a single micronutrient because higher nutrient levels were due to higher intake in diet and protein intake was a dietary factor that is preferentially restricted in CKD patients. We also performed stratified analyses by diabetes mellitus (DM), and hypertension (HTN) origins. DM was defined as the presence of a history of diabetes and fasting blood glucose ≥126 mg/mL at baseline. HTN was defined as a patient on antihypertensive medication, or the presence of a history of hypertension, systolic blood pressure ≥140 mm/Hg, and diastolic blood pressure ≥90 mm/Hg. 

In order to observe the effect of one micronutrient on the risk of CKD development under adjustment for multiple micronutrients and other risk factors, we constructed a clinico-nutritional model which was a full multivariable composite model, composed of two models, a dietary nutritional model and a clinico-epidemiological model. Each dietary nutritional model and clinico-epidemiological model was constructed by backward Cox regression models. We assessed the collinearity between independent variables using the variance inflation factor (VIF) and the Pearson’s correlation coefficient. The starting model of the dietary nutritional model included 6 minerals, vitamins E and vitamin C, retinol among the vitamin A and its precursors, and vitamin B2, folate, and vitamin B6, which were significant in the single-vitamin analysis among the vitamin B complex. By using the backward Cox model, zinc, potassium, vitamin E were deleted. Although total energy and protein intake were not statistically significant in univariable analysis, both were included in the multivariable nutritional model because higher nutrient levels were due to higher intake in diet and protein intake is a dietary factor that is preferentially restricted in CKD patients. The starting model of the clinico-epidemiological model was constructed with 10 clinico-epidemiological factors used for single-micronutrient analysis and the stratification variables DM and HTN. By using the backward Cox model, alcohol consumption, total cholesterol levels in blood, the urine albumin to creatinine ratio, and uric acid in urine were deleted. Thus, eight variables (age, baseline eGFR, sex, BMI, physical activity, cigarette smoking, DM, and HTN) in the clinico-epidemiological model and eleven nutritional variables (calcium, phosphorus, sodium, iron, retinol, vitamin B2, folate, vitamin B6, vitamin C, total calories and protein intake) in the dietary nutritional model were finally selected. Additionally, the clinico-nutritional model was composed of 19 variables. 

In addition, we evaluated the discriminatory accuracy of the dietary nutritional model, the clinico-epidemiological model and the clinico-nutritional model (a composite model of both models) in predicting CKD risk using Harrell’s C index. All statistical analyses were conducted using SAS version 9.4 (SAS Institute Inc., Cary, NC, USA). 

## 3. Results

### 3.1. General Characteristics

Compared with the non-CKD group, the CKD group was more likely to be older; female; of higher BMI; have a lower education level (below middle school); have lower protein/fat intake, serum hemoglobin, hematocrit, HDL cholesterol, ALT, total protein, albumin, urine creatinine, uric acid, sodium, and PH levels; and have higher serum total cholesterol, triglycerides, fasting blood sugar (FBS), glycated hemoglobin (HbA1C), high-sensitivity C-reactive protein (hsCRP), blood urea nitrogen (BUN), calcium, and potassium levels (Appendix A). 

### 3.2. Associations between Single-Mineral/Vitamin Intake and the Risk of Developing CKD

The associations with single-micronutrient intake on the risk of CKD stage 3A or stage 3B and over are shown in Table 1. When the mineral intake was divided into categories, the lowest and highest intake levels were associated with increased risk of CKD stage 3B and over compared with the reference levels, which included the RDA, AI, or National Kidney Foundation guideline values for calcium (<200 mg/day, HR = 2.0, 95% CI: 1.01–3.63), phosphorus (<400 mg/day, HR = 6.1, 95% CI: 3.22–11.58), iron (<7 mg/day, HR = 1.9, 95% CI: 1.22–3.10; ≥13.3 mg/day, HR = 2.1, 95% CI: 1.06–4.34), and zinc (5 mg/day, HR = 2.5, 95% CI: 1.23–5.13) (Table 1). 

When the vitamin intake was divided into categories, the lowest and highest intake levels were associated with increased risk of CKD stage 3B and over compared with the reference levels, which included RDA, AI, or National Kidney Foundation guideline values for retinol (<20 µg/day, HR = 2.2, 95% CI: 1.23–3.93), vitamin B2 (<0.7 mg/day, HR = 2.3, 95% CI: 1.18–4.45), folate (<100 µg/day, HR = 3.0, 95% CI: 1.04–8.39), vitamin B6 (<1.0 mg/day, HR = 2.7, 95% CI: 1.52–4.93; ≥1.6 mg/day, HR = 2.3, 95% CI: 1.12–4.66), and vitamin C (≥100 mg/day, HR = 2.3, 95% CI: 1.24–4.14) (Table 1). 

In addition, these results were consistent in the sensitivity analyses in the restricted cohort population (*n* = 8901) excluding newly defined CKD cases within 2 years from study entry (Appendix A). In the risk of diabetes and hypertensive CKD, although the strength of the association was slightly reduced due to the decrease in the number of cases, the direction of the association remained the same (Appendix A). None of the micronutrients showed a significant association with the risk of developing CKD stage 3A (Entire cohort, Table 1; Restricted cohort, Appendix A; For diabetes and hypertensive CKD 3A). The association between overall CKD risk and dietary factors was observed as a null effect. This was a result of reflecting CKD stage 3A because new cases being CKD stage 3A accounted for 88.2% out of all new CKD cases and the effect size of CKD stage 3A was similar to that of all cases (Table 1, Appendix A).

### 3.3. Discriminatory Accuracy of Models on Predicting the Risk of CKD 3B and over

Figure 2 shows the discriminant ability of each model to distinguish between new cases of CKD and non-cases over 12 year follow-up periods in the cohort population. With the ability to discriminate new CKD cases for the clinico-nutritional model, a multivariate model controlled by various micronutrients and other risk factors, the Harrell’s C-index at long-term follow-up was as follows: Model 1 (dietary nutritional) was constructed with 10 variables and the Harrell’s C-index was 0.7341. Model 2 (clinico-epidemiological) was constructed with 8 variables and the Harrell’s C-index was 0.9225. Model 3 (clinico-nutritional) was built using all variables in model 1 plus model 2 and the Harrell’s C-index was 0.9371. The clinico-nutritional model, a full-multivariable model, showed the highest differential accuracy among the three models predicting the risk of developing CKD stage 3B or higher. The C-index of the model was significantly different from that of the clinico-epidemiological model (*p* < 0.01).

The discriminant ability of the clinico-nutritional model including multiple micronutrients and risk factors to distinguish between new cases of diabetic and hypertensive CKD stage 3B and over and non-cases was also high (C-index 0.8989 and 0.9296, respectively) (Appendix A). We did not construct a clinico-nutritional model and a dietary nutritional model due to non-significant results in the association between any micronutrients and the risk of CKD stage 3A. 

### 3.4. Micronutrient Intake for the Risk of CKD Stage 3B and over in a Full Multivariable Clinico-Nutritional Model

Table 2 shows the adjusted HRs for the risk of CKD stage 3B and over associated with micronutrients in the full multivariable model (clinico-nutritional model) controlled by multiple micronutrients and other risk factors. Low phosphorus and vitamin B2 and high vitamins B6 and C intakes were positively associated with the risk of developing CKD stage 3B and over, respectively (phosphorus < 400 mg/day, HR = 6.78, 95% CI: 2.18–21.11; vitamin B2 < 0.7 mg/day, HR = 2.90, 95% CI: 1.01–8.33; vitamin B6 ≥ 1.6 mg/day, HR = 2.71, 95% CI: 1.26–5.81; vitamin C ≥ 100 mg/day, HR = 1.83, 95% CI: 1.00–3.33). Outside of the nutritional factors, age (HR: 1.17, 95% CI: 1.14–1.21), baseline eGFR (HR: 0.94, 95% CI: 0.93–0.95; *p* < 0.01), female (HR: 2.14, 95% CI: 1.21–3.79), smoking (HR: 1.96, 95% CI: 1.11–3.45), HTN (HR: 1.71, 95% CI: 1.23–2.38), DM (HR: 5.01, 95% CI: 3.51–7.15; *p* < 0.01) were shown to be statistically significant. Sensitivity analysis was performed in the restricted population excluding CKD cases that newly occurred within 2 years. An additional association with low folate levels (<100 µg/day) in higher risk of CKD stage 3B and over was observed (HR: 6.72, 95% CI: 1.40–32.16). Moreover, a much stronger association was observed with phosphorus intake. (<400 mg/day, HR: 8.51, 95% CI: 2.33–31.08; for 400–700 mg/day, HR: 2.42, 95% CI: 1.02–5.74). 

In addition, the association with the selected micronutrients and clinico-epidemiological factors on the risk of diabetic or hypertensive CKD stage 3B and over were persistent (Appendix A). None of the micronutrients showed a significant association with the risk of developing CKD stage 3A.

## 4. Discussion

In this study, we found that, among the general population, low dietary intakes of phosphorus, vitamin B2 (riboflavin), and folate and high dietary intake of vitamins B6 and C were associated with an increased risk of CKD stage 3B and over, compared with the intake at recommended levels in the multivariable Cox model adjusted for multiple micronutrients and other clinico-epidemiological risk factors. None of the categories of the minerals/vitamins showed a significant association with the risk of developing CKD stage 3A. 

Dietary restriction of phosphorus is recommended for CKD patients as a means of managing hyperphosphatemia because hyperphosphatemia is associated with kidney failure in CKD patients [17,18]. A prospective CKD patient cohort study suggested that hemodialysis patients (HD) must ingest no more than 800 mg/day of phosphorus [19]. It is correct to recommend low phosphorus intake for CKD patients. Our study group was not diagnosed with CKD at baseline and also had an eGFR of 60 or higher. When two of them were defined as a new CKD case two years later, phosphorus intakes <400 mg in the entire population (even phosphorus intakes <700 mg in the restricted population after excluding new CKD cases within 2 years) were associated with the risk of CKD stage 3B and over. Of CKD stage 3A patients, those with low phosphorus intake below 400 mg and below 700 mg accounts for 1.9% and 24.2%, respectively, but of patients being CKD stage 3B and over, those accounts for 9.2% and 34.6%, respectively. In the general population, low phosphorus intake means very low protein intake, i.e., malnutrition, which increases morbidity and mortality [20]. In our previous study, we also reported that low phosphorus intake (<664 mg; 1st quartile) increased the prevalence likelihood of CKD relative to 2nd quartile (664–844 mg) [10]. Low phosphorus intake may reflect that the main food of Koreans is rice and vegetables seasoned with salt and an eating habit of consuming less animal meat and dairy products.

Our previous study is the result of a study based on cross-sectional studies, so the reverse causation acts as a limitation. At the time the study was conducted, follow-up data were not disclosed to researchers. Therefore, despite the limitation of reverse causation in the cross-sectional study, the results were analyzed in the cross-sectional study setting [10]. In contrast, the current study presents causal association in the view of time relevance from exposure to outcome ascertainment because of cohort design showing the risk of CKD new cases. A trend study using repeated cross-sectional studies in Iran, which differs from the actual cohort study design, was conducted. According to that study, the lowest quintile level for phosphorus intake was 820.4 mg (average), and the probability of occurrence of CKD at the highest quintile level compared to the lowest quintile was found to have a null association [7]. We could not directly compare our results with other studies because there is no population that can observe very low phosphate intakes like ours. It is correct to recommend low phosphate intake for CKD patients. However, whether very low phosphate intake is indeed associated with the risk of developing CKD in the general population must be validated, thus requiring large-scale cohort studies involving people with low phosphate intake. 

Our results showed that the highest iron intake levels (<7 mg/day; ≥13.3 mg/day) were associated with an increased risk of CKD stage 3B and over compared with the intake at recommended levels (7–10 mg/day). CKD patients are recommended to consume adequate levels of iron to prevent anemia [21]. For the general population, there were few studies on the association between dietary iron intake and risk of CKD in prospective cohort study. Only a prior cross-sectional study in China showing that iron status was positive correlated with the odds of CKD supports our study result for iron [22]. Biologically, accumulation of iron in the urine or kidney is associated with the onset and progression of the kidney, and excessive iron May also increase the risk of inflammation and damage to free organs [23,24]. 

Vitamin B2 (riboflavin) status is not routinely measured in a healthy population. The body can store small amounts of riboflavin; therefore, urinary excretion reflects dietary riboflavin intake only after the tissues have been saturated [25]. Although there is no research proving that dietary riboflavin intake is associated with kidney function for the general population, two cross-sectional studies supported our results for riboflavin. The first study reported that low riboflavin and thiamine levels were associated with high fasting total homocysteine (tHcy) plasma levels in ESRD patients, and the second study reported a link between high tHcy levels and worse kidney function [26,27,28]. Vitamin B6 deficiency, like vitamin B2 deficiency, can lead to high levels of tHcy, which can worsen kidney function [28,29]. In a multivariable analysis corrected for several micronutrients, the risk of CKD for low B6 intake was observed as a null association, but high B6 intake was still associated with CKD risk. Prior a randomized controlled trial showing that excess vitamin B6 in patients with diabetic nephropathy resulted in a greater decrease in GFR supported our finding for high B6 intake [30]. Additionally, there was no cohort study of the general population directly supporting our findings for an increased risk of CKD at the lowest folic acid intake (<100 µg/day). A clinical trial reported that the simultaneous use of nalapril and folic acid delayed the eGFR deterioration compared to nalapril alone among CKD patients [31]. That result indirectly suggests that low folate levels are associated with high eGFR levels. Given the lack of direct evidence for vitamin B2, vitamin B6 and folic acid, further studies are needed on the association of these intake levels on the risk of developing CKD.

Prior results for vitamin C intake in relation to kidney disease were inconsistent across studies [7,32]. Our result showed that the highest vitamin C intake (≥100 mg/day) was associated with an increased risk of CKD stage 3B or higher compared to the recommended level (69–103 mg/day). The other cohort study that reported an increased risk of kidney stones compared to the recommended dose (<90 mg/day) with higher vitamin C intakes indirectly supported our results [32]. In contrast to our finding, a study from Iran reported the negative association between high intakes of vitamin C and CKD [7]. Although the epidemiological studies are controversial due to inconsistency, the association between high vitamin C intake and eGFR decrease is biologically plausible. Because metabolic by-products of ascorbic acid (i.e., oxalic acid deposits) can accumulate and reduce kidney function, it is recommended that CKD patients do not consume more than 100 mg of vitamin C per day [33]. 

In our general population, sodium intake is very high. This was the same in the results of a survey that could represent the sodium intake of the entire Korean population (Appendix A). High sodium intake in CKD patients can accelerate the progression of kidney disease or end-stage renal disease (ESRD) [34,35]. In CKD patients, it is clearly necessary to control sodium intake, but the association with CKD risk has not been clearly demonstrated in the general population of high-dose sodium intake, such as ours. The association between sodium and the risk of developing CKD in other general populations with low sodium intake should be established.

In the case of a single-nutrient analysis, low dietary intakes of calcium, phosphorus, zinc, and retinol were also observed to be associated with an increased risk of CKD. However, in the multivariable model, all of which were corrected for several nutrients, all of these nutrients were not statistically significant. This may be due to the fact that nutrients are consumed together from food, absorbing together or interacting biologically with other nutrients. In the case of zinc, when iron is included in the model, CKD risk and null association were observed, and it could be understood that they are related to each other. Biologically Iron and zinc correlated each other in the case of absorption and a zinc deficiency can coexist with an iron deficiency together [36]. 

When we analyzed the association with the risk of CKD development based on the average value of dietary factors, no association was observed. However, when we analyzed by classifying them into very low or very high categories and category near the RDA value, dietary intake was observed to be related to CKD risk. This suggests that there is a certain threshold in dietary intake in relation to CKD development and it can be associated with disease only if it is below or above the threshold.

Compared to the RDA of dietary intake, this cohort population tends to consume less calcium, potassium, vitamins A, E, and folic acid and higher sodium and vitamin C. This is the same for the general population (Appendix A). In addition, cohort members have a high proportion of people who consume less phosphorus and vitamin B2 (34.6% and 41.8% of all cases above stage 3B, respectively). Since the main dietary intake of Koreans is rice and vegetables seasoned with salt, many nutrients related to fat-soluble vitamins and protein intake tend to be low intake. The nutritional imbalance itself appears to be associated with the eGFR deterioration to CKD stage 3B and over.

Compared with the RDA value of dietary intake, calcium, potassium, vitamin A, E, and folate are ingested less, sodium and vitamin C are ingested a lot, and phosphorus and vitamin B2 are also tended to be low. This is the same for the general population (Appendix A). As the main dietary intake of Koreans is mainly vegetables including rice and salt, many nutrients related to the intake of fat-soluble vitamins and protein tends to be low intake. The nutritional imbalance itself appears to be associated with the eGFR deterioration to CKD stage 3B and over.

This study had several limitations. First, the definition of the incidence of CKD in this study may lead to overestimation of CKD patients. The definition of new CKD cases in this study was based on a case with decline of eGFR <60 after at least two years from the baseline, whereas the diagnostic definition of CKD stage 3A or higher is to find eGFR <60 in two consecutive measurements three months apart. Our definition based on a single measure may lead to overestimation of CKD in people with transient and reversible worsening of renal function, especially in early CKD. Our definition of CKD such as at least stage 3A CKD or higher by eGFR can result in subjects who may be stage 1 and 2 CKD in both the baseline and follow-up being included in the non-CKD population, leading to diluting the association. Our non-significant results for early CKD may be due to this limitation. The other limitation was that we could not use other biomarkers (i.e., the urine albumin to creatinine ratio, etc.) to ascertain new CKD cases since this cohort had only baseline information of biomarkers. Second, in the general population, people with temporary and reversible renal deterioration may limit dietary intake of some nutrients on the recommendation of a doctor, which may lead to reverse causation in our study results. To overcome this limitation to some extent, we additionally analyzed the restricted subjects who were excluded due to being new CKD cases that occurred within 2 years, and the results in the restricted subjects were confirmed to be consistent with those in the total subjects. Third, some dietary intakes in our population were at a very high (e.g., sodium) or very low level (e.g., calcium, potassium, vitamin A, and folate) beyond the recommended dose (RDA or AI). Nutritional deficiency may be due to low socioeconomic status (e.g., income, education levels). Thus, there may be limited generalizability for some results. Fourth, there is no information on dietary intake levels, suggesting the minimum risk level of CKD incidence among the general population to date. Although we considered a certain range at recommended intake levels (RDA or AI) as the minimum risk level, and categorized dietary intake levels according to increasing or decreasing a certain range, it is not clear whether the classification for our reference level is appropriate. Further studies on the appropriate intake levels confirming minimum CKD risk will be needed. However, this study has several strengths. The large sample size provided good statistical power to the results. As our results are for the association between the new cases of CKD development and dietary intake through 12 year follow-up observation of the general population, clinicians may be able to provide specific recommendations for patients based on these findings when planning nutrition management for people at risk of developing CKD. Nonetheless, the association with dietary intake in this study does not mean an association with exacerbation in patients with chronic kidney disease. In particular, it is obvious that high phosphorus intake for CKD patients should be restricted to prevent CKD progression. 

## 5. Conclusions

In conclusion, low dietary intakes of phosphorus, vitamin B2, and folate and high dietary intake of vitamins B6 and C were associated with an increased risk of CKD stage 3B and over, compared with the intake at recommended levels. There are few studies on the development of new CKD cases in normal subjects and there is sparse experimental and epidemiologic evidence for the association between CKD development and low or high dietary factors, including phosphorus intake, to date. Therefore, we could not find a plausible biological mechanism linking low phosphorus and high vitamin B6 intakes to increased CKD risk. Further investigations are required and should accumulate to elucidate the plausible mechanism that links dietary intake to CKD.

## Figures and Tables

**Figure 1 nutrients-13-01517-f001:**
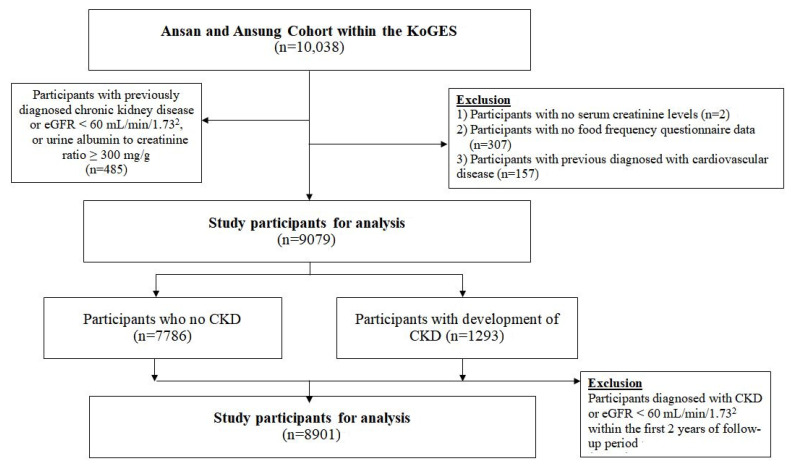
Study subjects to assess the association between dietary minerals and vitamins with development of chronic kidney disease, the Ansan-Ansung cohort of the Korean Genome and Epidemiologic study (KoGES). KoGES, Korean Genome and Epidemiologic Study; eGFR, estimated glomerular filtration rate; CKD, chronic kidney disease.

**Figure 2 nutrients-13-01517-f002:**
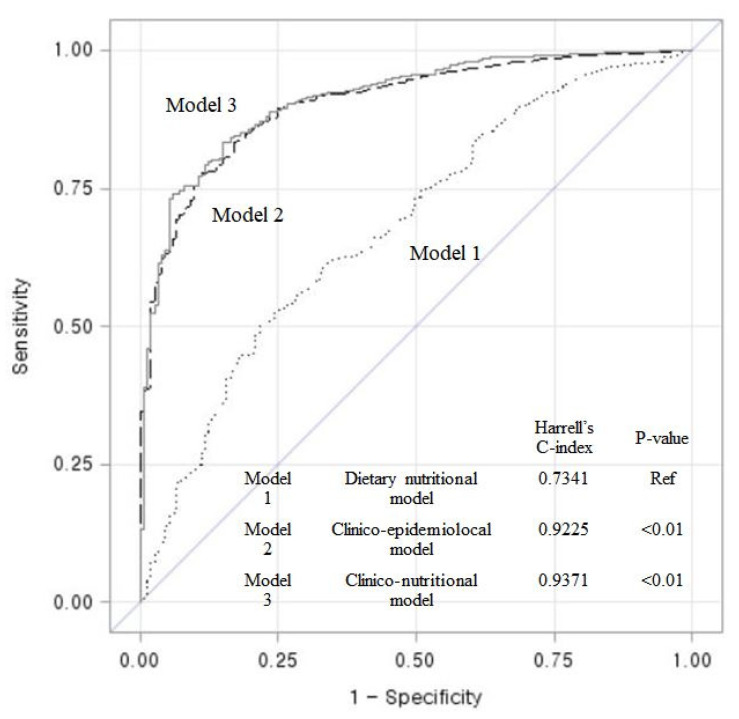
Receiver operating characteristic (ROC) curves and Harrell’s C-index showing the discriminant accuracy of each model for the ability to distinguish new cases of CKD stage 3B and over in the entire cohort population over 12 year follow-up periods. CKD, chronic kidney disease. Model 1 (dietary nutritional model): Function(Y) = β_1_[Calcium] + β_2_[Phosphorus] + β_3_[Sodium] + β_4_[Iron] + β_5_[Retinol] + β_6_[Vitamin B2] + β_7_[Folate] + β_8_[Vitamin B6] + β_9_[Vitamin C] + β_10_[Total Calories] + β_1__1_[Protein], Function(Y) = Log ($\frac{{\;\mathrm{Hazard}\;}_{\mathrm{Exposed}}}{{\mathrm{Hazard}\;}_{\mathrm{Non}-\mathrm{Exposed}}}$); Model 2 (clinico-epidemiological model): Function(Y) = β_1_[Age] + β_2_[Sex] + β_3_[Baseline eGFR] + β_4_[Physical activity] + β_5_[Cigarette smoking] + β_6_[Hypertension] + β_7_[Diabetes] + β_8_[Body mass index]; Model 3 (clinico-nutritional model, a composite model of Model 1 and Model 2): Function(Y) = $\beta_{11}^{1}$ [Model 1] + $\beta_{8}^{1}$ [Model 2]).

**Table 1 nutrients-13-01517-t001:** Single-mineral intake for CKD ^1^ risk in the entire cohort population (the Ansan-Ansung KoGES) with 12 year follow-up (*n* = 9079).

	Person-Years	CKD Stage 3A ^1^ (*n* = 1140)		Person-Years	CKD Stage 3B and over ^1^ (*n* = 153)		Person-Years	CKD (*n* = 1293)	
		Cases *n*	HR (95% CI) ^2^		Cases *n*	HR (95% CI) ^2^		Cases *n*	HR (95% CI) ^2^
Calcium (mg) ^3^									
<200	6176	124	1.0 (0.80–1.34)	5440	30	2.0 (1.01–3.62)	6322	154	1.1 (0.87–1.40)
200–400	27,123	465	1.1 (0.93–1.36)	24,181	52	1.0 (0.59–1.72)	27,394	517	1.1 (0.92–1.32)
400–600	22,004	295	1.0 (0.80–1.14)	20,158	37	1.0 (0.61–1.61)	22,207	332	1.0 (0.81–1.13)
≥600	19,102	256	1.0	17,462	34	1.0	19,293	290	1.0
Phosphorus (mg) ^3^									
<400	857	16	0.9 (0.52–1.41)	816	14	6.1 (3.22–11.58)	920	30	1.3 (0.90–1.90)
400–700	12,934	249	1.0 (0.84–1.18)	11,346	39	1.7 (1.04–2.65)	13,150	288	1.0 (0.88–1.21)
700–1200	40,450	617	1.0	36,646	64	1.0	40,789	681	1.0
≥1200	20,165	258	0.8 (0.70–1.01)	2151	36	0.9 (0.53–1.50)	20,356	294	0.9 (0.72–1.01)
Sodium (mg) ^3^									
<2000	16,601	297	1.0 (0.89–1.22)	14,682	40	1.2 (0.72–1.84)	16,808	337	1.0 (0.90–1.22)
2000–2999	21,936	332	1.0	19,797	34	1.0	22,099	366	1.0
3000–3999	17,540	239	0.9 (0.78–1.09)	15,973	33	1.1 (0.69–1.84)	17,714	272	0.9 (0.81–1.11)
4000–4999	9770	133	0.9 (0.75–1.14)	1049	22	1.4 (0.83–2.52)	9895	155	1.0 (0.80–1.17)
≥5000	8559	139	1.0 (0.84–1.28)	907	24	1.5 (0.87–2.71)	8700	163	1.1 (0.88–1.30)
Potassium (mg) ^3^									
<1400	8840	158	1.0 (0.76–1.30)	7903	35	1.5 (0.53–4.05)	9022	193	1.0 (0.79–1.29)
1400–2400	29,603	471	1.1 (0.86–1.32)	26,620	39	0.7 (0.30–1.57)	29,810	510	1.0 (0.81–1.20)
2400–3400	22,361	324	1.1 (0.93–1.35)	20,383	46	1.4 (0.70–2.20)	22,602	370	1.1 (0.93–1.32)
≥3400	13,602	187	1.0	12,335	33	1.0	13,782	220	1.0
Iron (mg) ^3^									
<7	14,913	272	1.0 (0.85–1.18)	13,250	48	1.9 (1.22–3.10)	15,164	320	1.1 (0.91–1.24)
7–10	21,142	340	1.0	18,986	31	1.0	21,303	371	1.0
10–15	26,076	369	1.0 (0.81–1.17)	23,763	44	1.5 (0.87–2.62)	26,315	413	1.0 (0.85–1.20)
≥15	12,274	159	0.9 (0.69–1.12)	11,241	30	2.1 (1.06–4.34)	12,434	189	1.0 (0.77–1.21)
Zinc (mg) ^3^									
<5	6484	143	0.92 (0.71–1.21)	5620	32	2.5 (1.23–5.13)	6637	175	1.0 (0.80–1.31)
5–8	30,145	478	0.9 (0.69–1.05)	27,221	57	1.3 (0.68–2.28)	30,453	535	0.9 (0.73–1.08)
8–11	23,234	339	1.0	21,044	38	1.0	23,448	377	1.0
≥11	14,543	180	0.8 (0.69–1.00)	13,355	26	0.9 (0.56–1.58)	14,678	206	0.8 (0.71–1.01)
Vitamin A (R.E) ^3^									
<300	19,067	368	1.1 (0.94–1.39)	16,828	54	1.0 (0.59–1.68)	19,333	422	1.1 (0.93–1.35)
300–500	24,116	360	1.1 (0.89–1.28)	21,814	38	0.7 (0.43–1.13)	24,309	398	1.0 (0.87–1.21)
500–700	14,441	198	1.1 (0.89–1.31)	13,124	24	1.0 (0.60–1.70)	14,567	222	1.1 (0.89–1.28)
≥700	16,783	214	1.0	15,475	37	1.0	17,006	251	1.0
Retinol (µg) ^3^									
<20	13,540	300	1.1 (0.86–1.31)	11,714	62	2.2 (1.23–3.93)	13,849	362	1.2 (0.95–1.40)
20–60	25,553	402	1.2 (0.96–1.39)	22,939	43	1.4 (0.79–2.46)	25,765	445	1.2 (0.99–1.41)
60–100	18,528	243	1.1 (0.90–1.32)	17,034	29	1.3 (0.72–2.34)	18,707	272	1.1 (0.93–1.34)
≥100	16,784	195	1.0	15,553	19	1.0	16,895	214	1.0
Carotene (µg) ^3^									
<1200	13,517	248	1.0 (0.82–1.23)	11,995	34	0.9 (0.53–1.58)	13,685	282	1.0 (0.83–1.21)
1200–2300	27,116	430	1.1 (0.91–1.27)	24,502	53	0.8 (0.54–1.33)	27,381	483	1.1 (0.90–1.24)
2300–3400	15,849	224	1.0 (0.87–1.26)	14,282	29	1.1 (0.67–1.81)	16,007	253	1.0 (0.88–1.25)
≥3400	17,924	238	1.0	16,461	37	1.0	18,144	275	1.0
Vitamin E (mg) ^3^									
<5	11,027	206	1.0 (0.81–1.33)	9817	39	1.1 (0.52–1.94)	11,209	245	1.0 (0.83–1.31)
5–8	22,751	400	1.3 (1.03–1.55)	20,213	35	0.7 (0.37–1.24)	22,950	435	1.2 (0.97–1.43)
8–11	19,223	271	1.1 (0.94–1.34)	17,543	40	1.0 (0.65–1.70)	19,429	311	1.1 (0.95–1.32)
≥11	21,404	263	1.0	19,667	39	1.0	21,628	302	1.0
Vitamin B1 (mg) ^3^									
<0.9	17,773	339	1.0 (0.85–1.18)	15,727	51	1.3 (0.79–1.96)	18,038	390	1.0 (0.87–1.19)
0.9–1.2	18,463	306	1.0	16,625	35	1.0	18,645	341	1.0
1.2–1.5	19,236	252	0.9 (0.77–1.14)	17,620	32	1.3 (0.71–2.34)	19,404	284	1.0 (0.79–1.15)
≥1.5	18,934	243	0.8 (0.66–1.04)	17,269	35	1.2 (0.58–2.42)	19,128	278	0.9 (0.69–1.06)
Vitamin B2 (mg) ^3^									
<0.7	17,674	361	1.0 (0.82–1.28)	15,478	64	2.3 (1.18–4.45)	18,004	425	1.1 (0.89–1.35)
0.7–0.9	18,651	294	1.0 (0.85–1.26)	16,859	31	1.4 (0.78–2.60)	18,811	325	1.1 (0.88–1.28)
0.9–1.2	19,248	263	1.0	17,502	25	1.0	19,381	288	1.0
≥1.2	18,833	222	0.9 (0.78–1.13)	17,401	33	1.2 (0.69–2.02)	19,021	255	1.0 (0.82–1.16)
Niacin (mg) ^3^									
<10	17,808	345	0.7 (0.58–1.00)	10,508	44	1.1 (0.58–2.28)	18,116	404	0.8 (0.61–0.98)
10–14	18,611	300	0.9 (0.69–1.06)	19,311	42	0.8 (0.44–1.50)	18,786	333	0.9 (0.70–1.06)
14–19	18,891	271	1.0	20,904	40	1.0	19,059	303	1.0
≥19	19,095	224	0.9 (0.76–1.09)	16,517	27	1.0 (0.57–1.60)	19,255	253	0.9 (0.77–1.09)
Folate (µg) ^3^									
<100	3555	57	1.1 (0.75–1.54)	3248	22	3.0 (1.04–8.39)	3663	79	1.2 (0.89–1.68)
100–200	25,961	433	1.2 (0.97–1.60)	23,253	47	1.1 (0.45–2.63)	26,224	480	1.2 (0.93–1.49)
200–300	26,083	396	1.2 (0.96–1.51)	23,536	36	0.7 (0.35–1.45)	26,255	432	1.1 (0.91–1.40)
300–400	11,099	153	1.2 (0.94–1.56)	10,177	28	1.4 (0.75–2.50)	11,244	181	1.2 (0.98–1.55)
≥400	7708	101	1.0	7027	20	1.0	7830	121	1.0
Vitamin B6 (mg) ^3^									
<1.0	5946	120	1.0 (0.80–1.25)	5292	30	2.7 (1.52–4.93)	6078	150	1.1 (0.91–1.37)
1.0–1.3	12,436	215	1.0 (0.86–1.25)	11,058	22	1.2 (0.65–2.25)	12,567	237	1.0 (0.87–1.24)
1.3–1.6	15,251	244	1.0	13,729	19	1.0	15,349	263	1.0
≥1.6	40,773	561	1.0 (0.82–1.20)	37,161	82	2.3 (1.24–4.14)	41,222	643	1.1 (0.89–1.28)
Vitamin C (mg) ^3^									
<60	13,034	215	0.9 (0.77–1.14)	11,815	30	1.4 (0.77–2.61)	13,176	245	1.0 (0.81–1.17)
60–75	8369	145	1.0 (0.84–1.29)	7459	17	1.7 (0.83–3.33)	8472	162	1.1 (0.87–1.31)
75–100	13,717	208	1.0	12,418	16	1.0	13,802	224	1.0
≥100	39,286	572	0.9 (0.79–1.11)	35,547	90	1.9 (1.07–3.22)	39,766	662	1.0 (0.85–1.17)

CKD, chronic kidney disease; KoGES, Korea Genome and Epidemiologic Study; eGFR, estimated glomerular filtration rate; CKD-EPI, Chronic Kidney Disease Epidemiology Collaboration. ^1^ CKD was defined as an eGFR <60 mL/min/1.73 m^2^ by the CKD-EPI criteria. Stage 3A and stage 3B and over were defined as 45 ≤ eGFR < 60 and eGFR < 45, respectively. ^2^ Cox proportional hazard model adjusted for age, sex, baseline eGFR, body mass index (BMI), regular physical activity, cigarette smoking, alcohol consumption, total cholesterol level in blood, hypertension, diabetes, the urine albumin to creatinine ratio, uric acid in urine, energy intake (mg/day), and protein intake (mg/day); for the analysis of ‘potassium intake’, additionally adjusted for folate intake/day. ^3^ For calcium, RDA (recommended dietary allowance for the general population) = 1300 mg; DRI (dietary reference intakes by KDOQI Clinical practice guideline for CKD patients or dialyzed CKD patients) = 800 mg (women CKD), 1000 mg (men CKD); for phosphorus, RDA = 700 mg; DRI = 400–700 mg (CKD); for sodium, AI (adequate intake for the general population) = 2000 mg; DRI = limit or reduce sodium intake (CKD); for potassium, AI = 3400 mg; DRI = adjust dietary potassium intake to maintain serum potassium within the normal range (CKD) and reduce dietary potassium intake (e.g., 2000 mg) (CKD 3–5 with hyperkalemia); for iron, RDA = 8 mg (women with age ≥51 or adult men), =18 mg (women with age 19–50); DRI = 10 mg (CKD). CKD patients need regular blood tests for iron overload; for zinc, RDA = 8 mg (women); DRI = 8 mg (women CKD); 10 mg (men CKD) 15 mg (dialysis); for vitamin A, RDA = 900 mg; DRI = 700–900 mg (CKD); for vitamin A precursor (retinol and carotene, not defined RDA or DRI); for vitamin E, RDA = 11 mg (women) 15 mg (men), DRI = more than RDA (CKD); for vitamin B1, RDA = 1.1 mg, DRI = 1.5 mg (CKD); for vitamin B2, RDA = 1.1 mg (women) 1.3 mg (men), DRI: 1.8 mg on a low-protein diet (CKD), 1.1–1.3 mg (dialysis), especially with poor appetite; for niacin, RDA = 14 mg (women) 16 mg (men); DRI = near to RDA (dialysis); for folate, RDA = 400 µg, DRI = more than RDA (CKD); for vitamin B6, RDA = 1.3 mg, DRI = 2 mg (dialysis); for vitamin C, RDA = 75 mg (women) 90 mg (men); DRI = 60–100 mg (dialysis).

**Table 2 nutrients-13-01517-t002:** Micronutrients and CKD ^1^ risk stage 3B and over in the full multivariable model (clinico-nutritional model) ^2^ controlling multiple nutrients and additional risk factors.

	Entire Cohort		Restricted Cohort ^3^	
	HR (95% CI) ^2^	*p*-Value	HR (95% CI) ^2^	*p*-Value
Age	1.17 (1.14–1.21)	<0.01	1.17 (1.14–1.22)	<0.01
Sex (Female)	2.14 (1.21–3.79)	<0.01	1.96 (1.06–3.64)	0.03
Baseline eGFR	0.94 (0.93–0.95)	<0.01	0.95 (0.94–0.94)	<0.01
Smoker	1.96 (1.11–3.45)	0.02	1.54 (0.83–2.84)	0.17
Physical activity	1.20 (0.85–1.68)	0.29	1.08 (0.12–2.05)	0.67
BMI (≥25)	1.27 (0.90–1.79)	0.17	1.19 (0.82–1.72)	0.36
Hypertension	1.71 (1.23–2.38)	<0.01	1.79 (1.24–2.56)	<0.01
Diabetes	5.01 (3.51–7.15)	<0.01	5.46 (3.71–8.03)	<0.01
Diet energy intake	0.94 (0.56–1.59)	0.82	0.94 (0.54–1.65)	0.83
Diet protein intake	1.41 (0.71–2.78)	0.32	1.43 (0.69–2.95)	0.33
Calcium (mg) ^4^				
<200	0.68 (0.21–2.20)	0.51	0.77 (0.21–2.85)	0.69
200–400	0.84 (0.34–2.05)	0.69	0.98 (0.37–2.60)	0.97
400–600	1.06 (0.54–2.08)	0.86	1.21 (0.58–2.55)	0.60
≥600	1.0	-	1.0	-
Phosphorus (mg) ^4^				
<400	6.78 (2.18–21.11)	<0.01	8.51 (2.33–31.08)	<0.01
400–700	1.89 (0.95–4.11)	0.09	2.42 (1.02–5.74)	0.04
700–1200	1.0	-	1.0	-
≥1200	0.62 (0.29–1.36)	0.23	0.60 (0.26–1.37)	0.22
Sodium (mg) ^4^				
<2000	0.95 (0.54–1.69)	0.86	0.95 (0.50–1.82)	0.88
2000–2999	1.0	-	1.0	-
3000–3999	1.18 (0.70–1.99)	0.54	1.14 (0.64–2.04)	0.65
4000–4999	1.25 (0.67–2.32)	0.47	1.45 (0.75–2.83)	0.26
≥5000	1.38 (0.69–2.76)	0.36	1.76 (0.84–3.68)	0.13
Iron (mg) ^4^				
<7	1.00 (0.46–2.17)	0.99	0.93 (0.41–2.15)	0.87
7–10	1.0	-	1.0	-
10–15	1.11 (0.58–2.13)	0.74	1.22 (0.60–2.51)	0.57
≥15	2.05 (0.72–5.82)	0.17	2.06 (0.66–6.46)	0.21
Retinol (µg) ^4^				
<20	1.78 (0.81–3.92)	0.15	1.78 (0.76–4.16)	0.18
20–60	1.28 (0.63–2.62)	0.49	1.24 (0.57–2.68)	0.59
60–100	1.25 (0.67–2.34)	0.47	1.33 (0.68–2.61)	0.40
≥100	1.0	-	1.0	-
Vitamin B2 (mg) ^4^				
<0.7	2.90 (1.01–8.33)	0.04	2.69 (0.95–8.10)	0.08
0.7–0.9	1.81 (0.87–3.74)	0.11	2.13 (0.98–4.64)	0.05
0.9–1.2	1.0	-	1.0	-
≥1.2	0.92 (0.43–1.95)	0.81	1.11 (0.49–2.55)	0.06
Folate (µg) ^4^				
<100	2.57 (0.66–10.08)	0.17	6.72 (1.40–32.16)	0.02
100–200	1.39 (0.48–4.04)	0.54	1.35 (0.44–4.19)	0.60
200–300	0.74 (0.33–1.70)	0.48	0.60 (0.25–1.44)	0.25
300–400	1.38 (0.71–2.70)	0.34	1.09 (0.53–2.23)	0.75
≥400	1.0	-	1.0	-
Vitamin B6 (mg) ^4^				
<1.0	0.90 (0.32–2.56)	0.84	0.43 (0.12–1.55)	0.19
1.0–1.3	0.87 (0.42–1.81)	0.71	0.97 (0.44–2.14)	0.93
1.3–1.6	1.0	-	1.0	-
≥1.6	2.71 (1.26–5.81)	0.01	3.08 (1.34–7.09)	<0.01
Vitamin C (mg) ^4^				
<60	0.64 (0.31–1.32)	0.23	0.45 (0.19–1.04)	0.06
60–75	1.39 (0.69–2.80)	0.35	1.45 (0.68–3.06)	0.33
75–100	1.0	-	1.0	-
≥100	1.83 (1.00–3.33)	0.05	1.79 (0.98–3.41)	0.08

CKD, chronic kidney disease; KoGES, Korea Genome and Epidemiologic Study; eGFR, estimated glomerular filtration rate; CKD-EPI, Chronic Kidney Disease Epidemiology Collaboration. ^1^ CKD was defined as an eGFR < 60 mL/min/1.73 m^2^ by the CKD-EPI criteria. Stage 3B and over was defined as an eGFR <45, respectively. ^2^ The clinico-nutritional model was a multivariable Cox proportional hazard model, constructed as Function(Y) = $\overset{n}{\underset{i}{\sum }}\beta i\left[\mathrm{all}\;\mathrm{variables}\right]$
which were listed in the table; Function(Y) = Log ($\frac{{\;\mathrm{Hazard}\;}_{\mathrm{Exposed}}}{{\mathrm{Hazard}\;}_{\mathrm{Non}-\mathrm{Exposed}}}$). ^3^ Restricted cohort analysis excluding new CKD cases within 2 years from cohort entry. ^4^ For calcium, RDA (recommended dietary allowance for the general population) = 1300 mg; DRI (dietary reference intakes by KDOQI Clinical practice guideline for CKD patients or dialyzed CKD patients) = 800 mg (women CKD), 1000 mg (men CKD); for phosphorus, RDA = 700 mg; DRI = 400–700 mg (CKD); for sodium, AI (adequate intake for the general population) = 2000 mg; DRI = limit or reduce sodium intake (CKD); for iron, RDA = 8 mg (women with age ≥51 or adult men), = 18 mg (women with age 19–50); DRI = 10 mg (CKD). CKD patients need regular blood tests for iron overload; for vitamin A precursor (retinol and carotene, not defined RDA or DRI); for vitamin B2, RDA = 1.1 mg (women) 1.3 mg (men), DRI: 1.8 mg on a low-protein diet (CKD), 1.1–1.3 mg (dialysis), especially with poor appetite; for folate, RDA = 400 µg, DRI = More than RDA (CKD); for vitamin B6, RDA = 1.3 mg, DRI = 2 mg (dialysis); for vitamin C, RDA = 75 mg (women) 90 mg (men); DRI = 60–100 mg (dialysis).

## Data Availability

Data described in the manuscript, code book, and analytic code will be made available upon request pending.

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
