# Peer review of "Dietary Micronutrients and Risk of Chronic Kidney Disease: A Cohort Study with 12 Year Follow-Up"

_nutrients, 2021, doi:10.3390/nu13051517_

Round 1

Reviewer 1 Report

Very interesting and well written. I have few comments to the content of the paper:

  1. The generally accepted defininion to diagnose CKD stage 3a or higher is to find eGFR< 60 in two consecutive measurements three months apart. Hence, diagnosis based on single measure is inappropriate since classifies many patients with transient and reversible worsening of renal function and having CKD. This is of special significance for CKD3a (right beow 60 ml/min./1.73m2). Although authors state in the limitations that the incidence of CKD may be underestimated for certain reasons, this approach may actually lead to its overestimation and should also be mentioned as important limitation.
  2. In my opinion all the statements concerning causality are quite 'risky'. It is generally accepted that high phosphate intake may promote the development and progression of CKD. I am quite convinced that an association between low phosphate intake and the risk of development of CKD stage 3a or higher may rather reflect protein intake restrictions in patients with borderline GFR (i.e. reduced but still above 60). The same may hold true for low intake of vitamins B2, B6, folate or retinol: reduced intake may reflect changes in dietary intake of nutrients. In Discussion section the authors mix these issues, for example on page 11, when they state that low folate intake may increase the risk of developing CKD, and in next phrase they write that low intake of folic acid may be due to adherence to dietary restrictions. In my opinion this hypothesis might be verified by the comparison of nutirient intake between patients who are unaware and aware of having decreased eGFR (but still above 60) at the start of the observation. 
  3. At the end of a day, the key issue is how to explain findings - what is the biological link between increased or decreased intake of any analyzed nutrient and the outcome (CKD progression). I must admit that in a discussion section authors do not provide satisfactory explanations for their findings. Comments regarding phosphate, iron, zinc or vitamins are unconvincing and largely speculative. For example, statement that iron deficiency may lead to escalation of ESA doses and ESA may be harmful is true, but is entirely not related to the study findings: patients at this stage of CKD never receive ESA, and ESA (although have many side effects) do not induce CKD progression. 
  4. I acknowledge the importance of the findings - but in my opinion although the results are important and interesting, their interpretation is wrong. I would suggest to re-write the paper focusing on associations rather that searching for cause-effect links. In my opinion the authors - leaving the results section as it is - should re-interpret their findings in the Discussion.  

Author Response

1) Authors response by reviewer's comments

2) Revised manuscript 

3) Revised supplementary tables

Reviewer 2 Report

The study was conducted in a formally correct manner. However, a review of the literature shows that a work based on the same database has shown very similar conclusions to those of this paper. https://pubmed.ncbi.nlm.nih.gov/29795052/

Here is part of the abstract of this paper: "A cross-sectional analysis investigated the association between mineral intake (calcium, phosphorus, sodium, potassium, iron, and zinc) and CKD using the Health Examinee (HEXA) cohort of the Korean Genome and Epidemiologic Study (KoGES). The present study suggests that an inadequate intake of some minerals may be associated with CKD occurrence in the general population."

The discussion in this new paper is quite dull and only presents a list of micronutrients in correlation with the risk of kidney disease. 
Statistical analysis: Some possible confounding factors were not considered. For example could age influence the correlations?

The authors should point out the differences of this paper from the reported one. The discussion should provide suggestions for clinical practice or for the development of new original research on the subject.  

Author Response

(The authors gave the same response as above.)

Round 2

Reviewer 1 Report

The really big work has been done to imporve the content and quality of this submission. No more criticism. 

Author Response

We express gratitude for the reviewer's comments. 

Reviewer 2 Report

The authors clarified all the points I had raised in the previous review.

Author Response

We express gratitude for the reviewer’s comments. 
